# SAR Image Ship Target Detection Adversarial Attack and Defence Generalization Research

**DOI:** 10.3390/s23042266

**Published:** 2023-02-17

**Authors:** Wei Gao, Yunqing Liu, Yi Zeng, Quanyang Liu, Qi Li

**Affiliations:** 1Department of Information and Communication Engineering, School of Electronic Information Engineering, East Campus of Changchun University of Science and Technology, 7089 Weixing Road, Changchun 130022, China; 2Department of Robotics, School of Electronic Information Engineering, East Campus of Changchun University, 6543 Weixing Road, Changchun 130022, China

**Keywords:** SAR image, ship target detection, adversarial attack, adversarial defence, generalization

## Abstract

The synthetic aperture radar (SAR) image ship detection system needs to adapt to an increasingly complicated actual environment, and the requirements for the stability of the detection system continue to increase. Adversarial attacks deliberately add subtle interference to input samples and cause models to have high confidence in output errors. There are potential risks in a system, and input data that contain confrontation samples can be easily used by malicious people to attack the system. For a safe and stable model, attack algorithms need to be studied. The goal of traditional attack algorithms is to destroy models. When defending against attack samples, a system does not consider the generalization ability of the model. Therefore, this paper introduces an attack algorithm which can improve the generalization of models by based on the attributes of Gaussian noise, which is widespread in actual SAR systems. The attack data generated by this method have a strong effect on SAR ship detection models and can greatly reduce the accuracy of ship recognition models. While defending against attacks, filtering attack data can effectively improve the model defence capabilities. Defence training greatly improves the anti-attack capacity, and the generalization capacity of the model is improved accordingly.

## 1. Introduction

Synthetic aperture radar (SAR) [1,2] is a high-resolution imaging radar developed with digital processing technology. It has good all-weather capabilities and perspective and has been widely used in military reconnaissance surveying and mapping. Many researchers had studied the application of remote sensing in the ocean [3,4,5] for sensing the marine environment, marine and coastal environment detection. SAR was widely used in marine monitoring: oil spill monitoring [6] and marine biodiversity observation [7]. Ships are necessary equipment for marine development, energy transportation, national defence construction and other activities. The most important marine activity component is ship monitoring. SAR can penetrate clouds and fog with unique advantages and is not limited by meteorological conditions. It is very suitable for data sources for ship detection.

A ground SAR detection system operates in space tracks. Researchers have developed high-performance deep learning algorithms and frameworks. They are widely used in SAR ship detection systems to solve practical problems. Although satellite communication has a high technical level, it is still vulnerable to security threats, which may seriously affect the judgement of the signal receiver. While the detection accuracy is important, we are more concerned with the relevant security risks of models. Neural networks have reached human performance on independent identically distributed testing machines [8,9,10]. However, it is difficult to distinguish an image that contains weak naked noise that the eyes cannot detect. This noisy image data, which have serious impacts on automatic detection systems, cause a detected object to deviate from the real object or a significant offset in the detection of anchor frames. Confrontation data [11] are slightly modified images. The purpose of the modification is to interfere with the result of a machine learning analyser. The model parameters can be modified by modifying the attack data without modifying the program. This attack will have a serious impact on a model, and it is very easy to cause misdetections, missed reports or false reports. SAR image ship-detection systems need to adapt to an increasingly complicated actual environment, and the requirements for the stability of detection systems continue to increase. The focus of this research is to ensure the stability of a system, prevent the system from being disturbed by external noise, and prevent a decrease in accuracy of a detection system when there is minor interference.

This article answers the following questions:What is the impact of protecting SAR ship models from attacks?How can stronger confrontation data be generated?Which attack form has the greatest impact on a SAR ship detection model?How does these model samples help the model improve the accuracy in the defence?

The structure of the paper is as follows: The second part introduces related work, and the third part studies the expansive adaptive gradient as well as its derivative formula. The fourth part evaluates the gradient expansion attack model. The fifth part is the conclusion.

## 2. Related Work

### 2.1. Adversarial Attack

The use of machine learning in smart networks brings potential security threats. The purpose of maliciously injected fake training data is to destroy the learning model. Due to the development of neural network adversarial attacks, researchers have conducted much research on offensive attacks. According to the degree of model consistency, they can be divided into black box attacks and white box attacks. Black box attacks [12,13] can only launch confrontation attacks by querying the output classification results of an input sample. White box attacks [14,15] use information, such as model structures, parameters and other information, to conduct confrontation attacks. Common attack methods include gradient information algorithms and interpreted algorithms based on neural networks. The target data set of a SAR image ship model is also easily affected by sample attacks. Wang et al. [16] applied the Momentum Iterative Fast Gradient Sign Method (MI-FGSM) and ADVGAN algorithms to SAR data sets to generate confrontation samples and conduct SAR image classification attacks. The experimental results show that confrontation samples are destructive for SAR image models.

### 2.2. Gradient-Based Attack Methods

In machine learning algorithms, when minimizing the loss function, a minimum loss function and the corresponding parameter values can be found through gradient decline. Conversely, to maximize the loss function, it can be found through gradient expansion. By disturbing a vector, it can be superimposed on a sample to form an attack sample. Through an attack, the output results are as large as possible with the deviation in the input.

In this section, we briefly introduce several gradient-based attack methods. A white-box attack is based on gradient-based optimization and restrained optimization. Among them, several types of gradient optimization have a greater impact.

The Fast Gradient Sign Method (FGSM) [13] is a basic method for white box attacks. This method quickly guides the model to find the direction in which the loss function increases:(1)$$x^{\prime }=x+\in \cdot sign\left(\nabla_{x}J\left(f\left(x\right),y\right)\right)$$

The Basic Iterative Method (BIM) [17] is one of many extensions of FGSM. A confrontational example generated by BIM is defined as follows:(2)$$x_{i+1}^{\prime }=x_{i}^{\prime }+\alpha \cdot sign\left(\nabla_{x}J\left(f\left(x_{i}^{\prime }\right),y\right)\right)$$
where $x_{0}^{\prime }=x$, α=ϵ/T, and *T* denotes the number of iterations.

TIM [18] uses Gaussian nuclear volume stairs and can be combined with MIM:(3)$$g_{i+1}=\mu \cdot g_{i}+\frac{W*\nabla_{x}J\left(f\left(x_{i}^{\prime }\right),y\right)}{\|W*\nabla_{x}J{\left(f\left(x_{i}^{\prime }\right),y\right)\|}_{1}}$$
(4)$$x_{i+1}^{\prime }\leftarrow Clip_{\epsilon }\{x_{i}^{\prime }+\alpha \cdot sign\left(g_{i+1}\right)\}$$
where *W* is the Gaussian kernel and * indicates the convolutional operator.

Dong et al. [19] proposed Momentum Iterative Fast Gradient Sign Method (MI-FGSM) and Nesterov Iterative Fast Gradient Sign Method (NI-FGSM). MI-FGSM integrates momentum into iterative attacks, thereby providing a higher transitability for confrontation examples. NI-FGSM improves the transplantability of confrontation examples. A Nesterov accelerated gradient can be integrated into an iterative gradient [20] basic attack to obtain a robust attack model.

This simple method is used to illustrate the success rate of using iteration methods in subsequent work. To generate malicious data that may be classified by a model, more iterative updates are needed. The calculation time and the number of iterations are linear, so more time is needed to create stronger attacks.

Later, the predictable perturbation attack (PPA) algorithm was used to add restricted indicators as the gradient rises.

### 2.3. Adversarial Defence

Studies have shown that deep learning networks have obvious weaknesses when processing data that contain noise. The disturbances contained in these pictures are intentional, very small, and cannot be perceived by human beings. This noise is difficult to detect by human eyes, but it has a serious impact on the test results. To solve this problem, researchers have designed many defence methods that focus on using good models trained on large data sets to correct adversarial examples. These methods include the adversarial training method [21,22,23], gradient regularization method [24] and method based on input transformations [25,26,27].

For example, in [28], the DeepFool algorithm was proposed to effectively calculate the disturbance cause by a deceptive deep network, thereby reliably quantifying the robustness of classifiers. A large number of experiments show that this method is better than existing methods in calculating improving a classifier’s robustness.

### 2.4. SAR Image Noise

SAR image noise is mainly of two types: additive noise and multiplication noise. This article focuses on the impact of additive noise on a SAR ship detection system. Additive noise usually uses zero average white noise as a model [29]. In the subsequent sections, we use the noise model that is the closest to the actual noise distribution, that is, normal distribution noise with a mean value of 0, to protect against attack noise.

## 3. Materials and Methods

### 3.1. Defending SAR Ship Data (NAA)

After a ship’s information is utilized, if the system is attacked, the hidden hazard information will bypass the defence model to attack the system, which will cause the ship’s detection model to fail and affect the test results.

There are three general approaches to defend against attacks during the model training phase: adjusting the training data, adjusting the labels, and adjusting the input features. It is more difficult for an attacker to attack model labels and input features. Adjusting the training data is the most obvious and most effective attack-response method. Hence, this article proposes a ship detection data attack model. Through these methods, the original distribution of the training data is attacked by injecting confrontation samples and adjusting, modifying or deleting training data. The adjusted data will cause the model to misjudge the detection results to achieve an attack effect.

The framework of this article consists of two parts: sensitive directional estimation and disturbance selection.

### 3.2. Sensitive Directional Estimation

Sensitive directional estimation refers to the opponent’s sensitivity to changes in the altered features of each input feature through the data stream around sample *X*.

### 3.3. Disturbance Signal Selection

Interference selection refers to the use of the characteristics of the information of a ship to select “interference” *σ* signals to obtain the most effective confrontation results. At the beginning of each new iteration, the modified sample *σ* + *X* replaces the original pure input sample *X* until the disturbance sample meets the opposition target conditions. Therefore, the goal is to determine a suitable disturbance. The total interference added to the original sample is as small as possible to achieve the goal of attacking the model while not being detected by the human eye.

The specific design ideas are as follows:

*X* is the input sample, and *f* is the model trained in this article. The goal of the opponent is to generate a confrontation sample $\hat{X}=\sigma +X$. In this process, a disturbance *σ* and input sample *X* are added. If the norm ||·|| describes the differences between the points in the input domain, creating a confrontation sample in model *f* can be formally turned into the following optimization problem:(5)$$X^{\prime }=X+argmin\left\{\left|\left(\left|\varphi \right|\right)\right|:f\left(\varphi +X\right)\neq f\left(X\right)\right\}$$

There are two reasons why f(σ+X)≠f(X). First, this nonequality allows the classification category error to be obtained, and it makes it possible for large-scale offsets to be performed in the detection box. φ is shifted in the direction of the highest sensitivity of *X* and can obtain the best disturbance effect on the basis of minimal disturbances. The noise adaptive attack algorithm is used in this article to determine the most suitable disturbance amount. In the early stage of the algorithm, an effective supervision sample needs to be found with the rise in the gradient.

The traditional AdaGrad method uses historical gradients. In this article, an attenuation coefficient is added at the cumulative square gradient stage to control how much of the historically submitted information is obtained. The gradient accumulation is transformed into the moving average of the decaying exponential parameter to optimize the degree of gradient utilization.

Suppose the initial parameter is *τ*; the training concentration contains a small batch of *m* samples $\left\{x^{\left(1\right)},\ldots ,x^{\left(m\right)}\right\}$, and the corresponding target is $y^{\left(i\right)}$. The gradient is calculated as:(6)$$V_{t}=\sum_{T-\tau }^{T}g_{\tau }^{2}$$
(7)$$\eta_{t}=\alpha \cdot m_{t}/\sqrt{V_{t}}$$
where *α* is the initial learning rate, $m_{t}=\phi \left(g_{1},g_{2},\ldots ,g_{t}\right)$ is used to calculate the first-order momentum of the historical gradients, and $g_{1}$ represents the first-order gradient.

The gradient optimization method of the adaptive learning rate is adopted here. It makes the parameter learning rate self-adaptive, performs large updates on nonfrequent parameters, and performs smaller updates on frequent parameters. Therefore, it is very suitable for processing sparse data. AdaGrad is more robust than SGD.

It is a second-order gradient summation for all moments, and it is improved later in this paper using a recent second-order gradient sum. The neural network is under nonconvex conditions, so the latter method will perform better in this experiment.

### 3.4. Attack Noise

The adaptive noise attack algorithm used in this paper adds a large amount of noise to the data, and this noise is normally distributed. The disturbance value Δ*x* is added to the FGSM algorithm to improve the gradient expansion mode. The disturbance value must be as small as possible. The loss function after disturbance value training is larger than that used by the gradient decrease method. At present, the model loss function is set to *L*. The θ value generated during gradient expansion is determined. It is necessary to find Δ*x* to attack the model and decrease its performance; then, the *L* value is increased. The gradient expands on X because only one attack sample does not have an obvious impact, so repeated iteration is needed. After completing this process, the model input sample is $\hat{X}$, and the iteration leads to $f\left(\hat{X}\right)\neq f\left(X\right)$.

The expression is as follows:(8)$${X^{\prime }}_{t}=X_{t-1}+lr*\frac{\partial L}{\partial x_{t-1}}$$
where lr denotes to the learning rate and ∂L/∂x is the gradient of the loss function. After this step, the attack sample $\hat{X}$ can be better supervised. The iteration process is optimized with AdaGrad.

Then, η ~ n (μ, Δ) to sample multiple noise types from the normal distribution, where *μ* = 0. This sampling method guarantees that the input sample $x^{\prime }$ and the original sample x after noise is added satisfy $E\left[\left|X^{\prime }-X\right|\right]\to 0$. Then, the performance of Δx is controlled by controlling the size of hyperparameter δ. Because the normal distribution of the selected sample is 0, the value of the noise to be distributed in the data sample after noise superposition is still stable. Then, ϵ needs to be constrained to ensure that the level of the formal difference in the normal distribution satisfies $∥x-x{∥}_{p}⩽\epsilon$. The added noise may not negatively impact the model, and much noise should be sampled and screened. The mathematical expression and its explanation are as follows:(9)η∼N(0,δ)

This formula indicates that the N-group variance is a normal distribution of δ averaged to 0. The square difference δ is adjusted according to the desired impact on ϵ. The highest noise $\eta_{t}$ that can maximize the loss function L(f(x+Δx),y) is selected. Its mathematical expression is:(10)$$\eta_{t}=arg\underset{\eta \in H}{max}\in L\left(f\left(I_{t-1}+\eta \right),y^{t}\right)$$
where η is the choice of disturbance noise that can maximize the value of *L*.
(11)$$I_{t}=H_{t-1}+\varphi \left(X^{\prime }-\left(H_{t-1}+\alpha \eta_{t}\right)\right)$$
where $I_{t-1}$ refers to the attack sample of t−1, and the algorithm continues attack training. In addition, $\hat{x}-\left(I_{t-1}+\alpha \eta_{t}\right)$ represents the difference in the vector dimensions of the other effective attack samples. The setting of super parameter *φ* implies the presence of smaller supervision, which can preserve the noise features to the maximum extent. However, this also signals a reduction in the ability to attack, so multiple iterations are needed to find effective attack data. After the selection is completed, the average value of $X^{\prime }$ and the noise need to be determined for our method.

Therefore, this article expands the gradient, and it performs multiple iterations to obtain effective noise to achieve the purpose of the model. In addition, a Gaussian distribution with a mean of 0 is selected to control the level of the noise disturbance. The Gaussian distribution based on the dynamics of ϵ is used to generate candidate interference $\sigma_{0}$. The samples select the $X^{\prime }$ value with the highest loss function value and supervise the model with that function.

Finally, the minimum noise can be obtained by combining the two steps in the above method.

As shown in Figure 1, with the blue arrow η and after confirming the effective attack sample $\hat{x}$, a new attack sample is created. The candidate disturbance is obtained after *t* secondary iterations.

## 4. Results

SAR ship data are extremely scarce because they are difficult to collect and interpret. For the public data set used in [30,31,32,33,34], the data scale is far smaller than that of the current popular deep learning data set. There are many data differences: the sizes of the images are different, the polarization methods are different, and the difference in the resolutions is large. In this article, SSDD [35,36,37,38,39,40,41] data sets with complete scenes and many data samples are used as the experimental data sets. In the following experiments, to control the variables and exclude the interference factors that affect the results, we choose to compare the algorithms on the same data set.

We previously designed a statistical adaptation loss function based on attention and SAR ship data enhancement. Based on this model, we conduct related experiments on the confrontation attack model.

Experimental procedure:

To test the generalized performance of the model, the overall data are divided into three parts, i.e., α%, 20% and (100-20-α)% subsets, where α ∈ (20,60). These three subsets are the training data, testing data, and verification data, respectively. In the experiment, to ensure balanced training data and verification data, α = 50.

Moreover, 20% of the data are the regular test data of the model.

The attack samples are selected and added to the α% training data. The mixed α% training data are entered into the model, and the model performs defensive training.

(100-20-α)% of the data are always unchanged, as is the case when verifying the performance of third-party data test models. The reason for this is that it is difficult to find third-party data with similar distributions, sizes and target sizes in the available data sets to use as appropriate verification data. Therefore, using some data in their original format as verification data is a way to adapt to the environment.

The first four sets of data in Figure 2 show that the attack effect directly leads to model misses and misunderstandings. In Figure 2, although the target is still in the enclosure, the detection box has shifted sharply. The detection box shift is also a manifestation of the decline in the accuracy of the detection model after the attack.

## 5. Data Analysis

### 5.1. Attack Experiment

Table 1 shows the attack effects of different attack methods.

To evaluate object detection models, the mean average precision (mAP) [41] is used. The mAP of the original data is 97.88 when there is no attack. The attack effect of random noise on the model is not obvious, but the mAP is slightly reduced, by 4.3%.

When our NAA method is used, the checking rate and recall rate are the lowest. Compared with the several other attack methods in the table, our attack method is better. Moreover, the disturbance rate is the lowest, and the results prove that the data after interference are very close to the original image. The success rate of the NAA offense is high, and the generalization performance is the best. Even when the attack power of NAA is not the highest, the performance of the model after the defence procedures are applied is very high.

It can be seen that FGSM is infinitely iterated and damaged. Compared with FGSM, NAA is weak. The success rate of our NAA model is slightly higher than that of several other attack methods. All the attacks cannot be seen with the naked eye. Our purpose is to explore the impact of attack noise on a model when the disturbance rate is very low. The high and low significance of the disturbance rate indicator is not significant, and the naked eye cannot distinguish the images when the disturbance rate is low. This attack method can be used improve the defence performance of detection models so that the overall defence success rate will not decrease significantly.

### 5.2. Defensive Experiment

Creating simple attack models is not our goal. Our goal is to find a way to resist these attacks. Therefore, a series of defensive experiments are conducted.

The previously mentioned defence experiments are used to show effectiveness of a defence model in combating attack data when they are masked in the training data. The defence effect obtained by the training model is shown in Table 2. Compared with the results in Table 1, it can be seen that the accuracy of the detection model after the defence procedure has improved significantly. The attack samples generated by the NAA method are significant. These samples are better than other attack samples for model defence training.

We are very interested in exploring the performance of the model before and after defence training. To understand the real situation, we performed a set of comparative experiments. Using the data division method mentioned earlier, the overall data are divided into three parts, α%, 20% and (100-20-α)%, where α∈(20,60). α% of the data are used as training data, 20% are used as test data, and the remaining (100-20-α)% are used as verification data. In the experiment, to ensure balanced training data and verification data, α=50.

Table 3 shows the performance of the model on the 20% test set after using the α% data to fight the attack. This is because of the scarcity of SAR ship data and the very large differences in public data. In general, the variables must be controlled, and the data changes must be objectively reflected. This makes the model perform poorly because the training data are insufficient. However, this does not affect the anti-defence effect of our observation and comparison models. The focus of our attention is the trend of the data change.

The data shown in Table 4 use the α% data training model to verify the detection effect on the (100-20-α)% data set. Attack-response training is not used for the verification data set; it is only used for testing, and the purpose is to compare it with the model after defence training.

For the results shown in Table 5, the α% data are used to train the models, and then the α% data are used as confrontation data for defensive training. The detection effect is verified on the (100-20-α)% data set.

Notably, in Table 3, Table 4 and Table 5, the experimental data designed for Table 4 and Table 5 can be used in multiple tests (cross-verification). In each experiment, the data are verified to ensure the objectivity and fairness of the results.

The experimental results show that compared with the defensive monitoring model, the mAP obtained by attack-sample defence training is improved.

It can be seen from the comparison of Table 4 and Table 5 that NAA confrontation defence training improves the detection performance on the third part of the data. In the comparison experiments, due to the decrease in the amount of training data, the overall data performance is lower than that in Table 1 and Table 2. However, from the overall data, it can still be seen that the generalization ability of the model is increased.

## 6. Conclusions

Here, we address the original questions of the study. Gaussian distributed noise with a preset average value of 0 is added to the original data as confrontation attack data. Compared with other random noise types, Gaussian distributed noise is more similar to actual noise data. The gradient expansion method is used to combine the noise to fight against attack methods generated by adapting the **attack** algorithm. This attack method has a strong negative effect on the SAR ship detection model, it can greatly reduce the accuracy of the identification results obtained by the ship model, and it has a lower disturbance rate. The attack data that are screened during training can effectively improve the defence capabilities of the model. The anti-defence ability of defence training and the generalization ability of the model are strong.

## Figures and Tables

**Figure 1 sensors-23-02266-f001:**
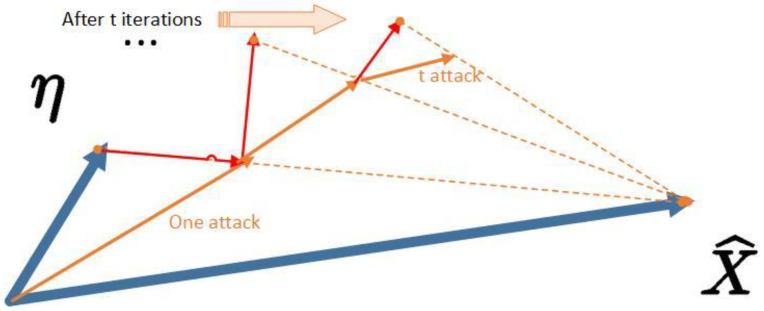
The η attack generation process.

**Figure 2 sensors-23-02266-f002:**
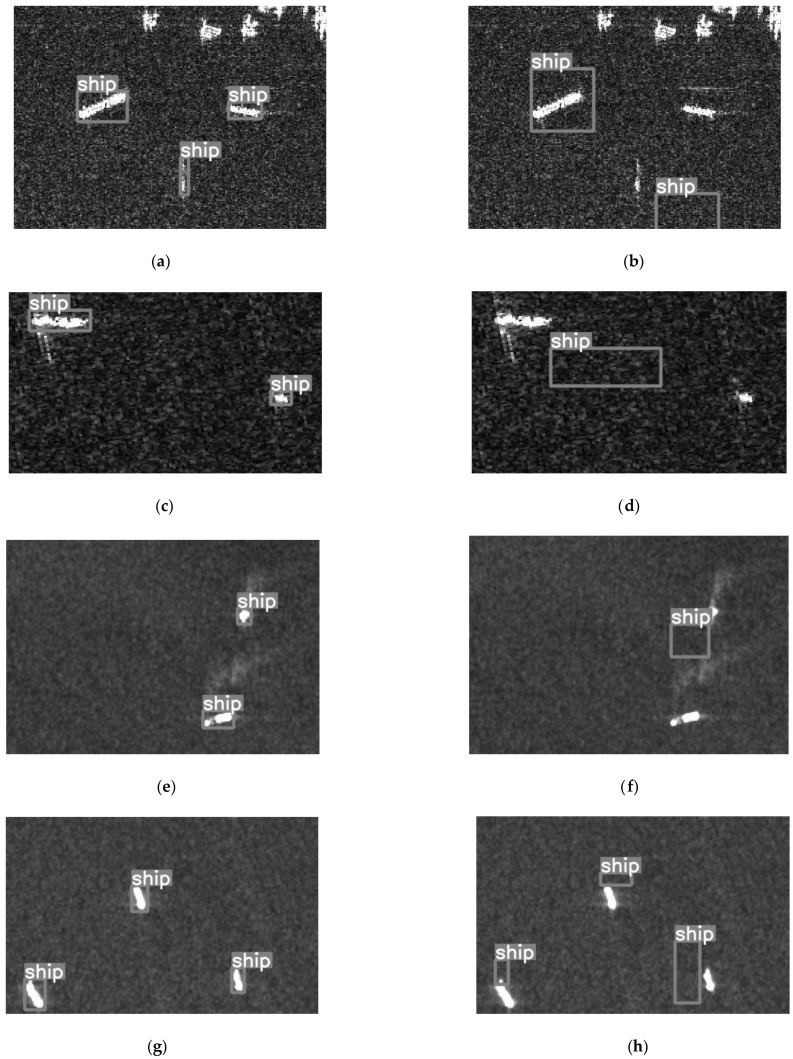
Comparison of the detection effect before and after the attack (The (**a**,**c**,**e**,**g**,**i**,**k**,**m**) panels show the detection effects of the models before the attacks. The (**b**,**d**,**f**,**h**,**j**,**l**,**n**) panels show the detection effects of the models after the attacks.).

**Table 1 sensors-23-02266-t001:** Comparison of the effects of different attack methods.

Attack	Precision	Recall	Success Rate	mAP
**Original**	97.02	97.62	0	97.88
**Random Noise**	93.12	93.36	6.2	93.58
**FGSM**	18.65	19.12	79.23	19.21
**AdvGAN**	16.53	17.12	80.18	17.32
**SI-NI**	15.52	15.68	82.69	16.12
**TIM**	14.32	14.65	84.76	14.87
**NAA**	11.27	11.83	82.5	11.86

**Table 2 sensors-23-02266-t002:** Defence effect comparison of different attack methods.

Attack	Precision	Recall	mAP
**FGSM**	95.28	95.72	95.94
**SI-NI**	95.87	96.28	96.42
**NAA**	96.21	96.71	96.82

**Table 3 sensors-23-02266-t003:** Test of the detection effect on the test sets when no defence training is applied.

Attack	Precision	Recall	mAP
**FGSM**	12.12	12.23	12.36
**SI-NI**	10.21	10.52	10.72
**NAA**	5.69	5.95	6.42

**Table 4 sensors-23-02266-t004:** Verification data test results when no defence training is applied.

Attack	Precision	Recall	mAP
**Original**	78.62	78.82	79.02

**Table 5 sensors-23-02266-t005:** Defence training verification test results.

Attack	Precision	Recall	mAP
**FGSM**	78.63	78.83	79.03
**SI-NI**	79.17	79.24	79.23
**NAA**	83.52	83.36	84.58

## Data Availability

The SSDD product used in this work is available at: https://github.com/TianwenZhang0825/Official-SSDD.

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
