# Peer review of "SAR Image Ship Target Detection Adversarial Attack and Defence Generalization Research"

_sensors, 2023, doi:10.3390/s23042266_

Round 1
Reviewer 1 Report
This paper presents the idea of defending SAR Image ship target detection from adversarial attack. It shows that the detection performance of SAR Image ship target detection can be damaged by adversarial attack and presents the idea how it can be mitigated. As this paper provides the attack and the defence algorithm and shows that their defence mechanism works with the existing attacks, I believe that the paper is meaningful for ones who study the ship target detection.
However, overall, the paper is a bit hard to read including the abstract. For example, in section 5, the meaning of mAP is not explained. Also, it is written differently in the tables (mAP) and in the explanation (MAP). As a non-expert in this area, it is hard to understand the analysis. Also, detailed explanation of the result is missing in the same section.
Author Response
Manuscript Number Sensors (ISSN 1424-8220)
Title: SAR Image Ship Target Detection Adversarial Attack and Defence Generalization Research
Dear Editor/Reviewers,
Thank you for all the comments. The following are our responses.
Reviewer #1:
This paper presents the idea of defending SAR Image ship target detection from adversarial attack. It shows that the detection performance of SAR Image ship target detection can be damaged by adversarial attack and presents the idea how it can be mitigated. As this paper provides the attack and the defence algorithm and shows that their defence mechanism works with the existing attacks, I believe that the paper is meaningful for ones who study the ship target detection.
- In section 5, the meaning of mAP is not explained.
Response: It has been done in the revised manuscript, as suggested.
- It is written differently in the tables (mAP) and in the explanation (MAP).
Response: It has been done in the revised manuscript, as suggested. mAP is a general evaluation index in the field of target detection. The full name of mAP is given in the paper. Because the derivation of mAP is too long, it is not put into the text of the paper. But we put the detailed derivation process in the reference literature.
All the changes are highlighted with red color in the revised manuscript.
Thank you for your work/help.
Sincerely,
Corresponding author:
Yunqing Liu, PhD
Changchun University of Science and Technology
7089 Weixing Road, Changchun 130022, China
Email: mzlyq@cust.edu.cn
Phone: +86 13843163761

Author Response
Manuscript Number Sensors (ISSN 1424-8220)
Title: SAR Image Ship Target Detection Adversarial Attack and Defence Generalization Research
Dear Editor/Reviewers,
Thank you for all the comments. The following are our responses.
Reviewer #1:
- The reference list should be improved. Although the introduction gives a good coverage of what has been done so far, there are some critical additions:
Within a sentence a more general remote sensing for the measurement of oceanic and marine parameters should be given:
- i) Synthetic aperture radar: Marine user’s manual, 2004.
- ii) https://unesdoc.unesco.org/ark:/48223/pf0000092618
iii) https://www.sciencedirect.com/science/article/pii/B9780128196045000019
- iv) https://ieeexplore.ieee.org/abstract/document/7954605
- v) https://www.mdpi.com/journal/jmse/special_issues/wl_remote_sensing
- vi) https://doi.org/10.5670/oceanog.2021.215
please enrich the introduction part by a few line paragraph addition.
Response: It has been done in the revised manuscript, as suggested.
- Please give full names of FGSM, BIM, TIM algorithms.
Response: It has been done in the revised manuscript, as suggested.
- Please mention that all subfigures in Fig.2 are ordered in a comparative basis. Left without attack, right with attack. With vs without attack noise. Also use small letters a,b,c,d.. in the caption to prevent confusion.
Response: It has been done in the revised manuscript, as suggested.
- Define mAP. In table it is given as mAP, in the text given as MPA!
Response: It has been done in the revised manuscript, as suggested.
- How does the proposed NAA algorithm work when noise is non-Gaussian? It is known that the behavior of noise effects the success of identification techniques such as SVM dramatically. Please provide a figure, similar to Fig2., but including the comparison of the results of the tested alogirthms vs proposed algorithm.
Response: All noises have an antagonistic enhancement effect on the model. This experiment is mainly aimed at Gaussian noise.
The actual situation is that neural networks can handle this noise. Usually, the data itself has noise, which may be various types of noise. In fact, noise enhancement is a very common data enhancement method. This data enhancement method has been proved to be effective. Use noise to enhance the data and let the neural network adapt to it. In the future, this kind of noise data can be matched by neural network.
The main idea of this paper is to use the gradient enhancement method to find the weak points of the model, so that the noise changes towards the more effective attack direction of the model, which is used to strengthen the anti-interference ability of the model.
All the changes are highlighted with red color in the revised manuscript.
Thank you for your work/help.
Sincerely,
Corresponding author:
Yunqing Liu, PhD
Changchun University of Science and Technology
7089 Weixing Road, Changchun 130022, China
Email: mzlyq@cust.edu.cn
Phone: +86 13843163761

Round 2
Reviewer 1 Report
Thank you for the authors to revised the paper, accordingly. I am happy with the current version. I only hope the authors to check the abstract. There are a hypen (e.g., algo-rithm) and white space (actual environment ,)
Author Response
Dear Editor/Reviewers,
Thank you for all the comments. The following are our responses.
There are a hypen (e.g., algo-rithm) and white space (actual environment ,)
Response: It has been done in the revised manuscript, as suggested.
All the changes are highlighted with red color in the revised manuscript.
Thank you for your work/help.
Sincerely,
Corresponding author:
Yunqing Liu, PhD
Changchun University of Science and Technology
7089 Weixing Road, Changchun 130022, China
Email: mzlyq@cust.edu.cn
Phone: +86 13843163761
Reviewer 2 Report
The authors addressed the majority of the corrections and additions.
Author Response
Thank you for your work/help.